# Rich nature of Van Hove singularities in Kagome superconductor CsV$_3$Sb$_5$

Yong Hu [1,12✉], Xianxin Wu [2,3,12], Brenden R. Ortiz [4,12], Sailong Ju [1], Xinloong Han [5,6], Junzhang Ma [7,8,9], Nicholas C. Plumb [1], Milan Radovic [1], Ronny Thomale [10,11], Stephen D. Wilson [4], Andreas P. Schnyder [2✉] & Ming Shi [1✉]

The recently discovered layered kagome metals AV$_3$Sb$_5$ (A = K, Rb, Cs) exhibit diverse correlated phenomena, which are intertwined with a topological electronic structure with multiple van Hove singularities (VHSs) in the vicinity of the Fermi level. As the VHSs with their large density of states enhance correlation effects, it is of crucial importance to determine their nature and properties. Here, we combine polarization-dependent angle-resolved photoemission spectroscopy with density functional theory to directly reveal the sublattice properties of $3d$-orbital VHSs in CsV$_3$Sb$_5$. Four VHSs are identified around the M point and three of them are close to the Fermi level, with two having sublattice-pure and one sublattice-mixed nature. Remarkably, the VHS just below the Fermi level displays an extremely flat dispersion along MK, establishing the experimental discovery of higher-order VHS. The characteristic intensity modulation of Dirac cones around K further demonstrates the sublattice interference embedded in the kagome Fermiology. The crucial insights into the electronic structure, revealed by our work, provide a solid starting point for the understanding of the intriguing correlation phenomena in the kagome metals AV$_3$Sb$_5$.

[1] Photon Science Division, Paul Scherrer Institut, CH-5232 Villigen, PSI, Switzerland. [2] Max-Planck-Institut für Festkörperforschung, Heisenbergstrasse 1, D-70569 Stuttgart, Germany. [3] CAS Key Laboratory of Theoretical Physics, Institute of Theoretical Physics, Chinese Academy of Sciences, 100190 Beijing, China. [4] Materials Department and California Nanosystems Institute, University of California Santa Barbara, Santa Barbara, CA 93106, USA. [5] Department of Physics and Center of Theoretical and Computational Physics, University of Hong Kong, Hong Kong, China. [6] Kavli Institute of Theoretical Sciences, University of Chinese Academy of Sciences, 100049 Beijing, China. [7] Department of Physics, City University of Hong Kong, Kowloon, Hong Kong, China. [8] City University of Hong Kong Shenzhen Research Institute, Shenzhen, China. [9] Hong Kong Institute for Advanced Study, City University of Hong Kong, Kowloon, Hong Kong, China. [10] Institute for Theoretical Physics, University of Würzburg, Am Hubland, D-97074 Würzburg, Germany. [11] Department of Physics and Quantum Centers in Diamond and Emerging Materials (QuCenDiEM) group, Indian Institute of Technology Madras, Chennai 600036, India. [12] These authors contributed equally: Yong Hu, Xianxin Wu, Brenden R. Ortiz. ✉email: yonghphysics@gmail.com; a.schnyder@fkf.mpg.de; ming.shi@psi.ch

Transition-metal based kagome materials, hosting corner-sharing triangles, offer an exciting platform to explore intriguing correlated[1-3] and topological phenomena[4-6], including quantum spin liquid[7-11], unconventional superconductivity[12-15], Dirac/Weyl semimetals[16-18], and charge density wave (CDW) order[1-5,13-15,19-25]. Their emergence originates from the inherent features of the kagome lattice: substantial geometric spin frustration, flat bands, Dirac cones, and van Hove singularities (VHSs) at different electron fillings. Recently, a new family of kagome metal $AV_3Sb_5$ (A = K, Rb, Cs)[26] with V kagome nets, was found to feature a $\mathbb{Z}_2$ topological band structure[27,28] and superconductivity was realized with a maximum $T_c$ of 2.5 K at ambient pressure[27]. Moreover, they exhibit CDW order below $T_{CDW} \approx 78$–103 K[29-31]. Aside from the translational symmetry breaking in this CDW phase, the breaking of additional symmetries, i.e., rotation and time-reversal, was observed upon cooling down towards $T_c$[29,32,33]. Despite evidences supporting a nodeless gap from magnetic penetration depth measurements[34], double superconducting domes under pressure[35-37], a large residual in the thermal conductivity[38] and an edge supercurrent in $Nb/K_{1-x}V_3Sb_5$ suggest electronically driven and unconventional superconductivity[39]. It is widely believed that these exotic correlated phenomena are intimately connected with the multiple VHSs in the vicinity of the Fermi level[40-42].

The characteristics of VHS bands are crucial in determining the Fermi surface instabilities[13-15,19-23]. From the perspective of band dispersion around the saddle point, VHSs can be classified into two types: conventional and higher-order[43,44], as shown in Fig. 1e (i) and (ii). The higher-order VHS displays a flat dispersion along one direction with less pronounced Fermi surface nesting, generating a power-law divergent density of states (DOS)

in two dimensions (2D) instead of a logarithmic divergent one[43,44]. Moreover, VHSs in kagome lattices possess distinct sublattice features: sublattice pure (p-type) and sublattice mixing (m-type), as shown in Fig. 1c. They induce an effective reduction of the local Coulomb interaction, thereby enhancing the role of non-local Coulomb terms[13,14,40]. Therefore, the nature of VHSs is pivotal to understand correlated phenomena, but still remains elusive in the kagome metals $AV_3Sb_5$ so far.

In this work, we perform a comprehensive study on the electronic structure of $CsV_3Sb_5$ by combining polarization-dependent angle-resolved photoemission spectroscopy (ARPES) measurements with density functional theory (DFT). The diverse nature of the four VHSs in the vicinity of the Fermi level ($E_F$) is directly revealed. We observe three VHSs around the M point below the $E_F$, formed by Vanadium $3d$ orbitals. Two of them are of conventional p-type, while the other one is of higher-order p-type. In addition, we find a conventional m-type VHS slightly above $E_F$ from our theoretical calculations. Furthermore, we show that the sublattice features are also embedded in the Dirac cone around the K point, exhibiting characteristic intensity modulations under various polarization conditions. Our study provides crucial insights into the electronic structure, thereby laying down the basis for a substantiated understanding of the correlation phenomena in the kagome metals $AV_3Sb_5$.

## Results

**Diverse nature of VHSs from theoretical calculations.** $CsV_3Sb_5$ crystalizes in a layered hexagonal lattice consisting of alternately stacked V-Sb sheets and Cs layers. Each V-Sb sheet contains a 2D vanadium kagome net interweaved by a hexagonal lattice of Sb atoms (Fig. 1a). The vanadium kagome lattice, shown in Fig. 1b,

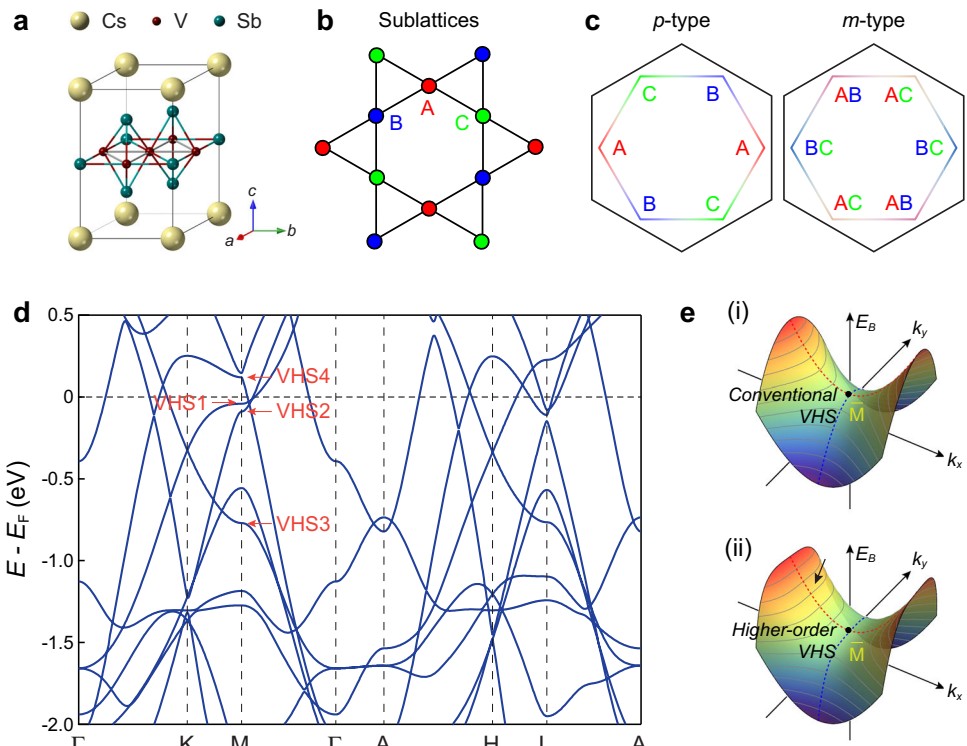

**Fig. 1 Crystal structure, Kagome sublattices and van Hove singularities in kagome superconductors $CsV_3Sb_5$. a** The Lattice structure of kagome metals $CsV_3Sb_5$. **b** Real space structure of the kagome vanadium planes. The red, blue, and green coloring indicate the three kagome sublattices. **c** Two distinct types of sublattice decorated van Hove singularities (VHSs) in $CsV_3Sb_5$, labeled as p-type (sublattice pure, left panel) and m-type (sublattice mixing, right panel). **d** Density functional theory calculated electronic structure of $CsV_3Sb_5$. The red arrows mark the VHSs. **e** Schematics of the conventional VHS (i) and higher-order VHS (ii) in two-dimensional electron systems. The gray curves in (**e**) indicate the constant energy contours that show markedly flat features along the $k_y$ direction in higher-order VHS, as highlighted by the black arrow.

hosts three distinct sublattices located at $3f$ Wyckoff positions. The toy band of the kagome lattice displays two different types of VHSs: p-type and m-type. For the p-type VHS, the states near the three M points are contributed by mutually different sublattices, while the eigenstates of the m-type VHS are equally distributed over mutually different sets of two sublattices for each M point, as illustrated in Fig. 1c. The band structure of $CsV_3Sb_5$ from DFT calculations is displayed in Fig. 1d, where four VHS points occur at M in the vicinity of $E_F$ (indicated by the red arrows and labeled as VHS1–4). Interestingly, VHS1 exhibits a much flatter dispersion along MK compared with the orthogonal direction MΓ. Further analysis shows that the quadratic contribution along MK is substantially reduced, indicating that VHS1 realizes a higher-order VHS (see Supplementary Fig. 4 for details). This is in contrast to the other three VHSs near $E_F$ which are all of conventional type with dominant quadratic dispersions in both directions. Motived by these observations, we perform ARPES measurements to investigate the electronic structure and focus on the nature of the VHSs in $CsV_3Sb_5$ which we infer from the polarization analysis.

**Multiple VHSs identified by ARPES.** The overall band dispersion and constant energy contours of $CsV_3Sb_5$ obtained via ARPES experiment are summarized in Fig. 2. The evolution of the electronic bands with different binding energy in Fig. 2a display

sophisticated structures, including Sb contributed electron pockets near the zone center and kagome-derived Dirac cones at the K points with binding energy ~0.27 eV (Fig. 2a and b). Photon energy-dependent measurement reveals a weak $k_z$ dispersion of the Vanadium $d$-orbitals (see Supplementary Fig. 1 for details), and we use the projected 2D BZ ($\bar{\Gamma}$, $\bar{K}$, $\bar{M}$) hereafter. Moreover, our temperature dependent measurement shows that the CDW order has some effect on the band dispersion around $\bar{M}$ point (see Supplementary Fig. 2). To reveal the VHS around $\bar{M}$ point, we display the band structure along the $\bar{\Gamma}$-$\bar{K}$-$\bar{M}$-$\bar{\Gamma}$ direction at 200 K in Fig. 2c, which are in good agreement with our DFT calculations (Fig. 2d). From the dispersion around $\bar{M}$ point, we clearly identify three saddle points, denoted by VHS1, VHS2 and VHS3. Particularly, VHS1 and VHS2 are just slightly below $E_F$ with strong intensity (See Supplementary Fig. 3 for details). Photon energy-dependent measurement suggests weak $k_z$ dispersion of the VHS2 band but moderate $k_z$ dispersion of the VHS1 band (see Supplementary Fig. 1). Remarkably, VHS1 exhibits a pronounced flat dispersion that extends over more than half of the $\bar{M}$-$\bar{K}$ path. Indeed, by fitting the experimental spectra we find that the quadratic term is substantially smaller than the quartic one, revealing the higher-order nature of VHS1 (Fig. 2e, see Supplementary Fig. 4 for details). Notably, the measured dispersion around VHS1 is much flatter than the theoretical one, indicating that renormalizations due to the electronic correlations

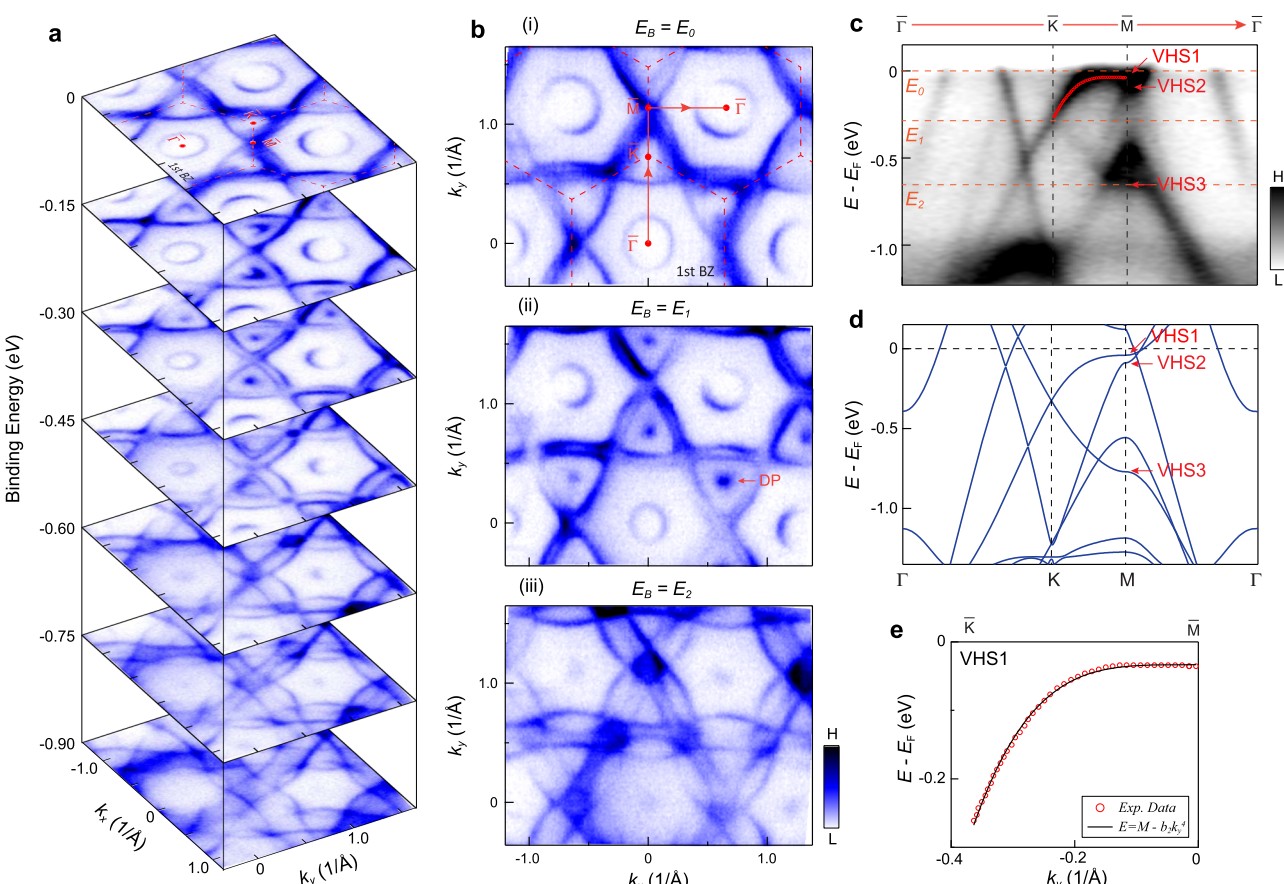

**Fig. 2 Identification of multiple van Hove singularities in CsV₃Sb₅. a** Stacking plots of constant energy contours at different binding energies ($E_B$) showing sophisticated band structure evolution as a function of energy. **b** Fermi surface (i), constant energy contours (CEC) at $E_B$ of the Dirac point (DP) (ii), and CEC at the VHS3 (iii). **c** Experimental band dispersion along the $\bar{\Gamma}$-$\bar{K}$-$\bar{M}$-$\bar{\Gamma}$ direction. The momentum direction is indicted by the red arrows in [**b**(i)]. The orange dashed lines indicate the energy position of the Fermi level, Dirac cone and VHS3. **d** Calculated bands along the $\bar{\Gamma}$-$\bar{K}$-$\bar{M}$-$\bar{\Gamma}$ direction. **e** Fittings of the measured dispersion along the $\bar{M}$-$\bar{K}$ by $E = M - b_2k_y^4$ form (the black line). The red dots represent the experimental data shown in (**c**) (see Supplementary Fig. 3 for the details). The red arrows in (**c, d**) mark the multiple VHSs. All measurements were probed with circularly polarized light, at 200 K, and the $\bar{\Gamma}$-$\bar{M}$ direction of the sample was aligned with the analyzer slit.

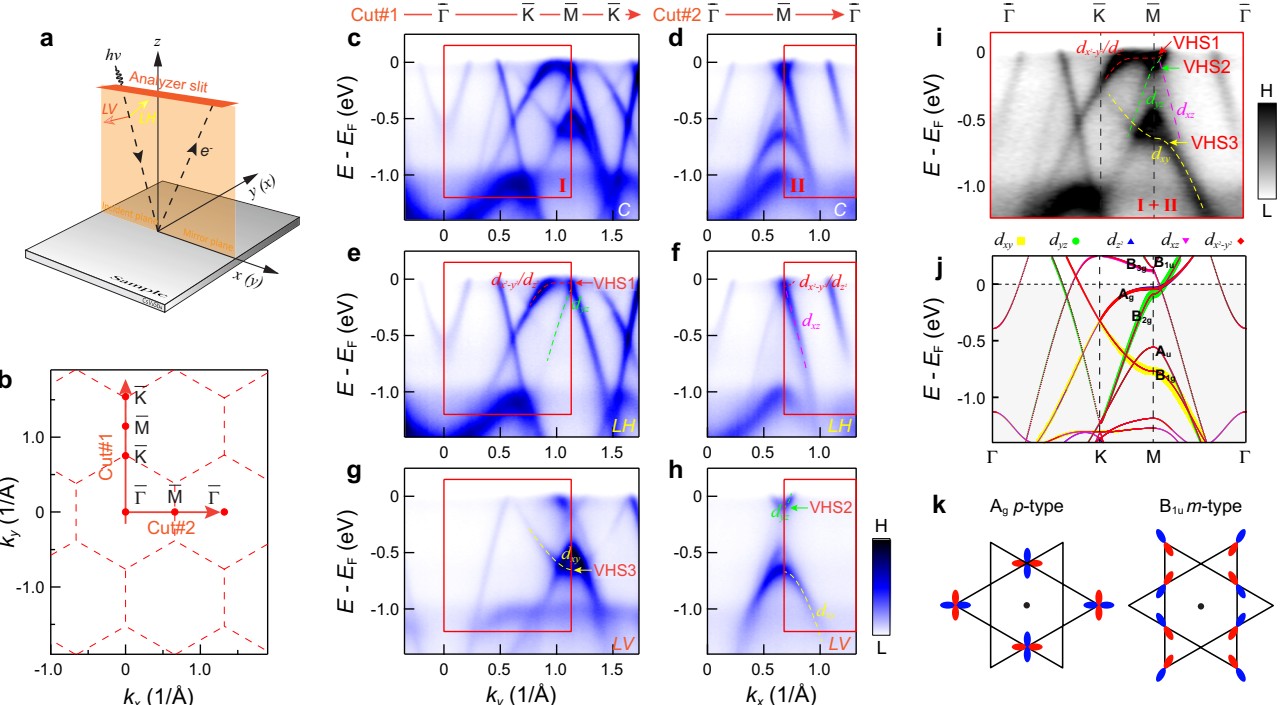

**Fig. 3 Determination of the orbital nature the kagome bands in CsV₃Sb₅. a** Experimental geometry of our polarization-dependent ARPES. **b** The two-dimensional projection of the Brillouin zone and the high-symmetry directions. **c, d** Band dispersions along the $\bar{\Gamma}$-$\bar{K}$-$\bar{M}$-$\bar{K}$ [Cut#1, (**c**)] and $\bar{\Gamma}$-$\bar{K}$-$\bar{\Gamma}$ [Cut#2, (**d**)] directions, respectively. The momentum directions of the cuts are indicated by the red arrows in (**b**). The bands were measured with circularly polarized light, at 200 K. **e, f** and **g, h** Same as (**c, d**), but probed with linear horizontal (LH) (**e, f**) and linear vertical (LV) (**g, h**) polarizations, respectively. We note that the intensity of the flat-top dispersion around the $\bar{M}$ point is weakened in (**e**), which may be due to the matrix element effects (see Supplementary Fig. 7 for the details of the matrix element analysis for the higher-order VHS band). **i** Experimental band structure along the $\bar{\Gamma}$-$\bar{K}$-$\bar{M}$-$\bar{\Gamma}$ direction, with orbital characters marked. The momentum range is equal to the sum of the region I in (**c**) and region II in (**d**) selected by the red box. The dispersion is the same as the cut shown in Fig. 2c. **j** Orbital character resolved band structure from the A sublattice in the calculations, with irreducible band representations labeled. **k** The sign structure (blue/red) and spatial orientation of the $d_{x^2-y^2}$-orbital $A_g$ p-type (inversion-even) and $d_{xz}$-orbital $B_{1u}$ m-type type (inversion-odd) VHS. The phase orbitals are plotted in the positive $k_z$ plane. The black dots in the center of the hexagons indicate the inversion centers.

enhance the higher-order nature of VHS1 (see Supplementary Fig. 4).

**Orbital and sublattice characters of the VHSs.** After directly identifying the VHSs in CsV₃Sb₅ from the band dispersion, we now turn to determining the sublattice nature of VHSs from the orbital symmetries by employing polarization-dependent ARPES measurements (see Supplementary Fig. 5 for experimental details). According to the selection rules in photoemission, bands can be selectively detected depending on their symmetry with respect to given mirror planes of the geometry[45]. Specifically, even- (resp. odd-) parity orbitals with respect to a mirror plane will be detected by the polarization whose electric field vector is in (resp. out of) the mirror plane. Our ARPES geometry is sketched in Fig. 3a. Polarization-dependent measurements were performed on the band structures along two orthogonal paths, $\bar{\Gamma}$-$\bar{K}$-$\bar{M}$-$\bar{K}$ and $\bar{\Gamma}$-$\bar{M}$-$\bar{\Gamma}$ directions (Fig. 3b). We first employ circular polarization that can detect both even- and odd-parity orbitals to map out the full dispersions, as shown in Fig. 3c, d. To determine the orbital character of the bands, we further adopt linear horizontal (LH) polarization (Fig. 3e, f) and linear vertical (LV) (Fig. 3g, h) polarization. The electric field vector of LH and LV polarized light is in and out the mirror plane, respectively. Thus, when aligning the $\bar{\Gamma}$-$\bar{M}$ direction of the sample to along the analyzer slit, $d_{xz}$, $d_{z^2}$ and $d_{x^2-y^2}$ are all of even symmetry with respect to the mirror plane and are detectable in the LH geometry. However, $d_{yz}$

and $d_{xy}$ are odd with respect to the mirror plane, and thus photoemission signals are only detectable in the LV geometry. Likewise, when the slit is along the $\bar{\Gamma}$-$\bar{K}$ direction, the orbitals can be selected accordingly (see Supplementary Table I). Based on these selection rules (see supplementary Fig. 5 for details of the matrix element analysis), the orbital characters of the bands constituting the VHSs below $E_F$ can be clearly identified, as shown in Fig. 3i. The VHS1, VHS2, and VHS3 are attributed to $d_{x^2-y^2}/d_{z^2}$, $d_{yz}$ an $d_{xy}$ orbitals, respectively. We note that the flat top of the VHS1 band exhibits relatively weak intensity along the $\bar{K}$-$\bar{M}$ direction in the LH polarization (Fig. 3e), while the full flat dispersion of the VHS1 can be observed at other photon energies (54, 108 eV, corresponding to $k_z = 0$ as same as 78 eV), indicating that the diminished intensity of the flat-band top at 78 eV (Fig. 3e) is attributed to the matrix elements effect (see Supplementary Fig. 7 for details).

**Discussion**

With the experimental determination of the orbital character around the $\bar{M}$ point, we plot the theoretical orbital-resolved band dispersion in Fig. 3j originating from sublattice A (Fig. 1b), which is invariant under mirror reflection $M_{xz}$ and $M_{yz}$. Comparing the orbital characters around M point in Fig. 3i, j, we find a good agreement between the experimental and theoretical results. Furthermore, the states at VHS1, VHS2 and VHS3 at $\bar{M}$ point, characterized by $A_g$, $B_{2g}$ and $B_{1g}$ irrep.,

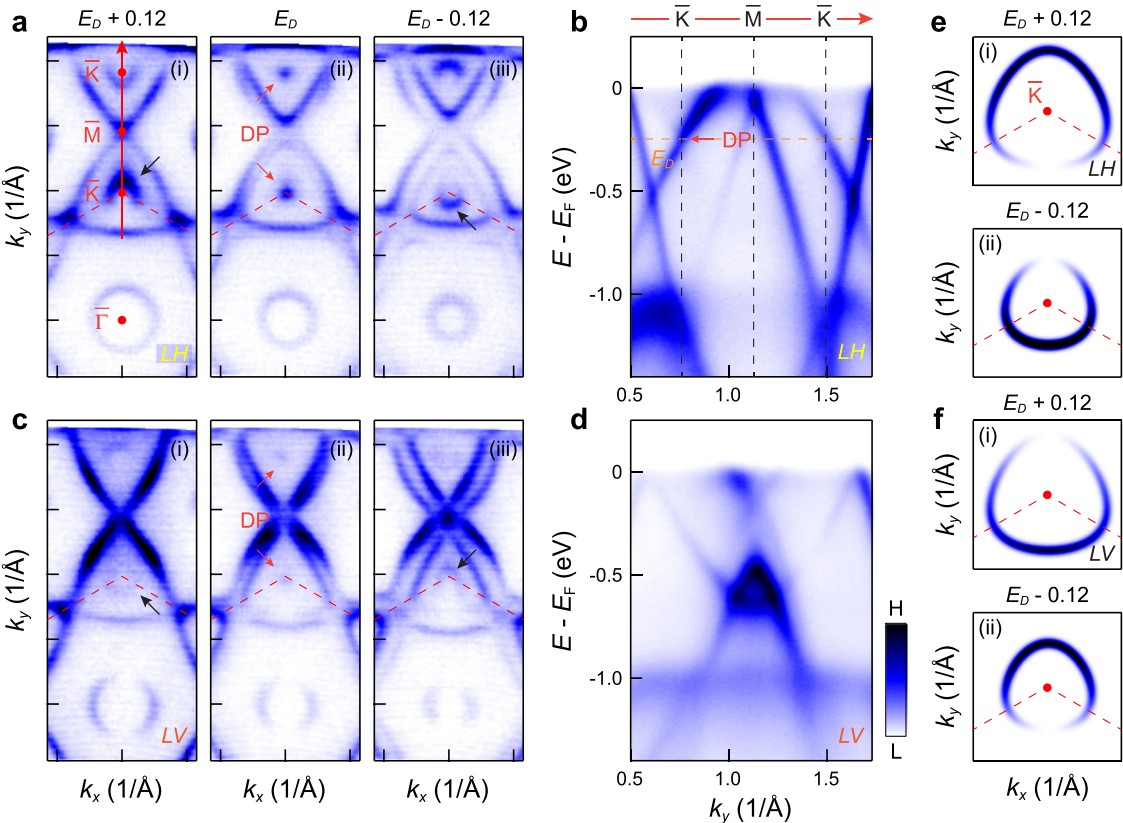

**Fig. 4 Modulation of the photoemission intensity around the Dirac point in CsV₃Sb₅. a** Constant energy maps above (i), at (ii) and below (iii) the Dirac cone, respectively. The black arrows indicate the spectra intensity pattern of the Dirac cone. **b** Experimental band dispersion along the K̄-M̄-K̄ direction. The data in (**a**) and (**b**) were probed with LH polarization. **c**, **d** Same as (**a**),(**b**), but measured with the LV polarization. All the experimental data (**a**–**d**) were obtained at 200 K. **e** Simulation of the constant energy map above (i) and below (ii) the Dirac cone for the LH polarization, based on the sublattice interference of kagome initial states. **f** Same as (**e**), but for the LV polarization. The red dashed lines in (**a**, **c**) and (**e**, **f**) show the Brillouin zone.

are solely attributed to the *A* sublattice and inversion-even, confirming their p-type nature (Fig. 3k). The band top of $d_{xz}$ band at M̄ is above $E_F$ and beyond experimental observation (Fig. 3i, j). However, our calculations show that this band belongs to $B_{1u}$ irrep. (inversion-odd, Fig. 3k) and is attributed to a mixed contribution from sublattices *B* and *C*, implying its m-type nature (see Supplementary Fig. 6). The four p-type and m-type VHSs, especially, the three of them (the VHS1, VHS2 and VHS4) are close to the Fermi level, suggesting that these VHSs with their large DOS and nontrivial sublattice and higher-order natures play a key role in driving the exotic correlated electronic states in AV₃Sb₅.

We further discuss the sublattice features of the Dirac cone bands around the K̄ point. The polarization dependent intensity patterns of the Dirac cones can convey the phase information of electronic wave functions, providing a way to determine the chirality of the Dirac cone[6,46,47]. Figure 4a displays representative constant energy contours of CsV₃Sb₅ around the K̄ point. The spectral intensity is strongly modulated around the kagome-derived Dirac cone for the LH polarization (Fig. 4b), with the maximum and minimum along the Γ̄-K̄ direction but at opposite momentum direction above [Fig. 4a(i)] and below [Fig. 4a(iii)] the Dirac point [Fig. 4a(ii) and b]. Similar behavior is observed in LV polarization but in a reversed fashion (Fig. 4c, d). The intensity modulation around the Dirac cone, mimicking the case of graphene[47] and the kagome metal FeSn[6], indicates the chirality of the kagome-derived Dirac fermions in CsV₃Sb₅. The spectral intensity patterns (Fig. 4a, c) can be excellently reproduced in a spectral simulation based on sublattice interference of kagome

initial states (Fig. 4e), further illustrating the sublattice interference embedded in the band structure of CsV₃Sb₅.

Our ARPES measurements, combined with DFT calculations, reveal the different natures of the four 3*d*-orbital VHSs near $E_F$ and the chiralities of the kagome-derived Dirac cones in CsV₃Sb₅. These are general features in the family of kagome metals AV₃Sb₅ and have important physical implications. For example, the nontrivial sublattice texture of the p-type VHSs leads, via sublattice interference, to a suppression of local Hubbard interactions and promotes the relevance of non-local Coulomb terms[13,40]. The bands from the conventional p-type VHS2 feature a good Fermi surface nesting, and the nesting vector connects parts of the Fermi surfaces dominated by different sublattices, which can lead to a 2 × 2 bond CDW instability. This could provide a reasonable explanation for the observed CDW order[48,49]. However, the origin of the CDW order is still under debate and phonons can play an important role[30]. The higher-order p-type VHS1, on the other hand, exhibits less pronounced Fermi surface nesting with large DOS, which could promote a nematic order[43,44], providing a possible explanation for the additional crystal symmetry breaking in the CDW phase at lower temperature. The appearance of multiple types of VHSs near $E_F$ including both p-type and m-type, derived from the multi-orbital nature, can induce a rich competition for various pairing instabilities and thus generate numerous different orders depending on small changes in the electron filling[40–44]. The kagome metals AV₃Sb₅ offer the tantalizing opportunity to access and tune these orders via carrier doping or external pressure[35,36,50–54], which remains to be further investigated both experimentally and theoretically.

## Methods

**Single crystals growth**. Single crystals of $CsV_3Sb_5$ were synthesized using the self-flux method. All sample preparation is performed in an argon glovebox with oxygen and moisture <0.5 ppm. The flux precursor was formed through mechanochemical methods by mixing Cs metal (Alfa 99.98%), V powder (sigma 99.9%), and Sb beads (Alfa, 99.999%) to form a mixture which is ~50 at.% $Cs_{0.4}Sb_{0.6}$ (near eutectic composition) and 50 at.% $VSb_2$. Note that prior to mixing, as-received vanadium powders were purified in-house to remove residual oxides. After milling for 60 m in a pre-seasoned tungsten carbide vial, flux precursors are extracted and sealed into 10 mL alumina crucibles. The crucibles are nested within stainless steel jackets and sealed under argon. Samples are heated to 1000 °C at 250 °C/h and soaked for 24 h before dropping to 900 °C at 100 °C/h. Crystals are formed during the final slow cool to 500 °C at 1 °C/h before terminating the growth. Once cooled, the crystals are recovered mechanically. Samples are hexagonal flakes with brilliant metallic luster. Elemental composition of crystals was assessed using energy-dispersive X-ray spectroscopy (EDS) using an APREO-C scanning electron microscope.

**ARPES measurements**. The samples were cleaved in situ with a base pressure of better than $5 \times 10^{-11}$ Torr. Angle-resolved photoemission (ARPES) measurements were performed at the ULTRA endstation of the Surface/Interface Spectroscopy (SIS) beamline of the Swiss Light Source. Data was acquired with a Scienta-Omicron DA30L analyzer using 78 eV photons with a total energy resolution of 18 meV. The Fermi level was determined by measuring a polycrystalline Au in electrical contact with the samples.

**Computational methods**. We employed first-principle calculations based on the density-functional theory (DFT) as implemented in the VASP[1–3]. The core electrons were treated using the projector-augmented wave method, and the exchange correlation energy was described by the generalized gradient approximation (GGA) using the PBE functional[4]. The cutoff energy for expanding the wave functions into a plane-wave basis was set to 500 eV. The Brillouin zone was sampled in $k$ space within the Monkhorst–Pack scheme[5] with a $k$ mesh of $9 \times 9 \times 5$, which achieved reasonable convergence of electronic structures. We have also done the calculations with spin–orbit coupling (SOC) and there are no qualitative changes except some gap openings, compared with the band without SOC. Therefore, we present the band structure without SOC in the main text to clearly identify the orbital characters. For all calculations, we used the experimentally determined crystal structure (space group P6/mmm, $a = b = 5.4949$ Å, $c = 9.3085$ Å).

## Data availability

The data that support the findings of this study are available from the corresponding authors upon reasonable request.

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

## Acknowledgements

The authors wish to thank J.-F. He for helpful discussions. The work was supported by the Swiss National Science Foundation under Grant. No. 200021-188413, and the NCCR MARVEL, a National Centre of Competence in Research, funded by the Swiss National Science Foundation (grant number 182892). The work at UC Santa Barbara was supported via the UC Santa Barbara NSF Quantum Foundry funded via the Q-AMASE-i program under award DMR-1906325. This research made use of the shared facilities of the NSF Materials Research Science and Engineering Center at UC Santa Barbara (DMR-1720256). B.R.O. acknowledges support from the California NanoSystems Institute through the Elings Fellowship program. Y.H. was supported by the National Natural Science Foundation of China (12004363). J.Z.M. is supported by the National Natural Science Foundation of China (12104379), Guangdong Basic and Applied Basic Research Foundation (2021B1515130007). M.R. acknowledges the support of SNF Project No. 200021-182695. R.T. is funded by the DeutscheForschungsgemeinschaft (DFG, German Research Foundation) throughProject-ID 258499086 - SFB 1170 and through the Würzburg-DresdenCluster of Excellence on Complexity and Topology in Quantum Matter - ct.qmat Project-ID 390858490 - EXC 2147.

## Author contributions

Y.H. and M.S. designed the research. B.R.O. grew and characterized the crystals with guidance from S.D.W. X.W. and X.H. performed the theoretical calculations with the support from A.P.S. and R.T. Y.H. performed the ARPES experiments with help from S.J., N.C.P., J.Z.M., M.R. and M.S. Y.H. analyzed the data and discussed with N.C.P. Y.H. draw the figures with help from X.W. and M.S. Y.H. and X.W. wrote the paper with inputs from A.P.S. and M.S. All authors contributed to the discussions. M.S., A.P.S., and Y.H. supervised the project.

## Competing interests

The authors declare no competing interests.
