## [Peer Review File · Nature Communications]

Editorial Note: This manuscript has been previously reviewed at another journal that is not operating a transparent peer review scheme. This document only contains reviewer comments and rebuttal letters for versions considered at Nature Communications

Reviewers' comments:

Reviewer #2 (Remarks to the Author):

In the reply letter, the authors made serious efforts to address my comments in previous report. In particular, the authors include the new photon energy dependence data at 200K with higher energy photons in the supplementary materials. However, the added photon energy dependence data are only focused at the zone center. While they are useful/sufficient for determining the periodicity in the k_z direction and estimating the inner potential, the critical information on whether there is substantial k_z dispersion of the VHS at M is still missing. In this regard, the unpublished data I mentioned in previous report are now posted on the web (arXiv:2105.01689v2). In Fig. S5 of this paper, the photon energy dependence data taken along K-M-K direction clearly show that the VHS discussed in current manuscript exhibit substantial k_z -dependence: it is located above E_f at 97 eV and below E_f at 106 eV. Note that all data in this figure were taken in LH polarization. At 106 eV, the VHS related band shows a band top below E_f with no suppression of intensity as expected; whereas at 97 eV, the corresponding band loses its intensity around M simply because the VHS shifts above E_f , without resorting to complex matrix element effect of dx^2-y^2 and dz^2 orbitals. For this reason, I cannot recommend the current manuscript for publication in its present format without a satisfactory resolution to the issue of k_z dispersion of VHS at M.

Point-to-Point Response to Reviewers' Reports

For clarity, the reviewers' original comments are shown by blue italic characters.

The authors' responses are shown by black normal characters.

Reviewer #2 (Remarks to the Author):

In the reply letter, the authors made serious efforts to address my comments in previous report. In particular, the authors include the new photon energy dependence data at 200K with higher energy photons in the supplementary materials. However, the added photon energy dependence data are only focused at the zone center. While they are useful/sufficient for determining the periodicity in the k_z direction and estimating the inner potential, the critical information on whether there is substantial k_z dispersion of the VHS at M is still missing. In this regard, the unpublished data I mentioned in previous report are now posted on the web (arXiv:2105.01689v2). In Fig. S5 of this paper, the photon energy dependence data taken along K-M-K direction clearly show that the VHS discussed in current manuscript exhibit substantial k_z -dependence: it is located above E_f at 97 eV and below E_f at 106 eV. Note that all data in this figure were taken in LH polarization. At 106 eV, the VHS related band shows a band top below E_f with no suppression of intensity as expected; whereas at 97 eV, the corresponding band loses its intensity around M simply because the VHS shifts above E_f , without resorting to complex matrix element effect of dx^2-y^2 and dz^2 orbitals. For this reason, I cannot recommend the current manuscript for publication in its present format without a satisfactory resolution to the issue of k_z dispersion of VHS at M.

We thank Reviewer #2 for reviewing our paper again and for providing further comments. We are glad to read that the Reviewer is satisfied with our k_z (thus *inner potential*) determination: “*In the reply letter, the authors made serious efforts to address my comments in previous report. ... they are useful/sufficient for determining the periodicity in the k_z direction and estimating the inner potential*”. The remaining concern is related to the k_z dispersion of VHS1 band and the flat-top dispersion of VHS1. In the following, we give our response to Reviewer #2's comments.

Following the Reviewer's suggestion, we display the photon energy-dependent data taken along the K - M - K direction with linear horizontal (LH) polarization at 200 K in Figs. R1. The VHS bands generally exhibit weaker k_z dispersion than the DFT calculations, and the evolution of the VHS1 band, as it crosses from below to above E_f , is clearly observed (see the black arrow in Fig. R1a), consistent with version 2 of the arXiv preprint 2105.01689v2. Moreover, the flat dispersion of VHS1 band is consistently seen in different $k_z = 0$ planes (as indicated by the orange dashed curve and black arrow in Fig. R1a), despite minor differences in spectral intensity. Notably, the k_z evolution of the VHS1 band is consistent with the k_z periodicity determined from the Sb p_z band around Γ , i.e.,

54 eV, 78 eV and 108 eV all correspond to the $k_z = 0$ plane, unambiguously confirming our previous estimate of the inner potential.

Importantly, we note that the flat feature of VHS1 in 78 eV (also presented in our main text) is relatively weaker compared to the 54 eV and 108 eV data (marked as the vertical arrow in Fig. R1a). On the other hand, the intensity of the VHS1 band below E_F in 78 eV is much stronger than the corresponding band in the 54 eV and 108 eV data (highlighted by the red horizontal arrows in Fig. R1a). We attribute this difference to matrix element effects, that depends on electron momentum, and on the energy and polarization of the incoming photon [1]. This interpretation is supported by our data collected with 78 eV *LH* polarization along a different momentum path ($\Gamma_1 - K - M, \Gamma_2 - K - M$ direction, indicated by the red arrow in Fig. R2a). This data shows that the VHS1 band has a clear nondispersive flat-top dispersion without any suppression of intensity, in contrast to the data along the $\Gamma - K - M$ direction (marked as the black arrow in Fig. R2a, compare Fig R1a with Fig R2b). In conclusion, in order to reveal the true dispersion of the VHS1 band one needs to take into consideration the $k_z = 0$ spectra measured at 54, 108 and 78 eV and also look at different momentum cuts. Hence, we conclude that the flat feature of the VHS1 band in the 78 eV spectra measured along the $\Gamma_1 - K - M$ direction, with *LH* polarization, is affected by matrix elements. This is in agreement with our previous theoretical calculations, where we included the matrix element effects to quantitatively explain the diminished intensity of the flat band top in 78 eV data.

Fig. R1 | Photon energy-dependent measurements on CsV₃Sb₅. **a** Photoelectron intensity plots along the $\Gamma - K - M - K$ path, measured with linear vertical (*LH*) polarization at 200 K. **b** Photon energy-dependent spectra along the $\Gamma - M$ direction, collected with circular polarization at 200 K.

Fig. R2| The flat feature under LH polarization. **a** Fermi surface map. **b** Experimental band structure along the $\Gamma_1 - K - M$ and $\Gamma_2 - K - M$ directions. The red arrow represents the momentum path, as indicated in (a). The black arrow in (a) indicates the momentum path ($\Gamma - K - M$) of the data shown in Fig. R1(a) and Fig. 3e in the main text. The data is measured with 78 eV, linear horizontal polarization (LH), and at 200 K.

Fig. R3| k_z dispersion of the VHS2 bands. **a** Photon energy-dependent spectra along the $\Gamma - K - M - K$ path (as indicated by the red arrow in the leftmost panel), measured with linear vertical (LH) polarization at 200 K. **b** Adapted from Fig. S5 of the preprint [arXiv:2105.01689v2]. The dashed line in (a,b), with the same height and shape, is guides to the eye for the VHS2 band **c** DFT calculated bands along $K - M - K$. **d** Summary of the guides to the eye from the plots shown in (b).

Regarding the k_z dispersion of VHS2, we did not observe noticeable k_z dispersion of the VHS2 bands (Fig. R3a), suggesting the k_z dispersion is weaker than the one of the DFT calculations. Actually, the k_z dispersion of VHS2 bands shown in the preprint [arXiv:2105.01689v2] is also much weaker than the DFT calculations. Figure R3b shows the measured k_z dispersion of the VHS bands in the preprint. Compared to the calculations (Fig. R3c), the experimental k_z dispersion of the VHS2 band is very weak (Fig. R3b). For instance, the VHS2 band (see the dashed line) shows a negligible difference in the $k_z = 0$ and $k_z = \pi$ planes (Fig. R3d), in sharp contrast to the DFT calculations (Fig. R3c). These results are fully consistent with our measurements.

Fig. R4 | The flat-top feature in CsV₃Sb₅. **a,b** Experimental band structure along the Γ - K - M path, measured with (a) circular polarization (C) and (b) linear horizontal polarization (LH), respectively, at 200 K. **c** Zoom-in plot of the bands along the MK direction in a selected region in (b), as indicated by the red box, and the Fermi-Dirac function is divided out to reveal the energy region slightly above E_F .

In summary, we appreciate the constructive comments and suggestions from the Reviewer. Regarding the Reviewer's concern, we have followed the Reviewer's suggestion and modified our paper accordingly:

(i) In the second round, as suggested by the Reviewer, we have invested considerable efforts to determine the k_z (thus *inner potential*), which has been recognized by the Reviewer.

(ii) Following the Reviewer's suggestions in the second round, we displayed the spectrum divided by the Fermi-Dirac function and demonstrated that there is no additional band crossing E_F along the M-K direction (Fig. R4). In Fig. R2, we have shown new data taken with LH polarization along a different path (namely, along the Γ_2 - K - M direction). This clearly shows that there is no additional band crossing E_F and that the VHS1 band below E_F has a flat band top without any suppression in intensity.

(iii) Furthermore, we provided additional photon energy-dependent measurements (Fig. R1) to further clarify the k_z dispersion of the VHS bands. In agreement with the preprint arXiv:2105.01689v2 (mentioned by the Reviewer), we find that the VHS1 has indeed a nonzero dispersion and crosses from below E_F to above E_F .

All of these results suggest that the observed relatively weak intensity of the flat-top dispersion around the M point in the 78 eV data can be attributed to matrix element effects, which is also supported by our previous theoretical calculations (Fig. S7). We would like to point out that this relatively weak intensity in the 78 eV data is a side issue and does not affect our main conclusion of the diverse nature of the VHSs in CsV₃Sb₅. Indeed, this does not affect our experimental finding of the sublattice texture and higher-order nature of the VHS. Our main conclusions are fully consistent with the second version of preprint [arXiv:2105.01689v2]. We hope that we have addressed all the technical concerns from the Reviewer and he/she will appreciate our efforts towards publishing these timely and important results in *Nature communications*.

References

[1] A. Damascelli, Z. Hussain, and Z.-X. Shen, Angle-resolved photoemission studies of the cuprate superconductors, *Rev. Mod. Phys.* **75**, 473–541 (2003).

Summary of Modifications

In the revised manuscript, the modifications are highlighted with red characters.

1. As suggested by Reviewer #2, we have added new data to show the k_z dispersion of the VHSs band, and updated Figs. S1 and S7 in the Supplementary Materials.

2. Considering the Reviewers' comments, we have mentioned the observed k_z dispersion of VHS1 band and the intensity variation in different $k_z = 0$ plane in the main text: "*Photon energy-dependent measurement suggests weak k_z dispersion of the VHS2 band but moderate k_z dispersion of the VHS1 band (see Supplementary Fig. S1).*", "*We note that the flat top of the VHS1 band exhibits relatively weak intensity along the $K - M$ direction in the LH polarization (Fig. 3e), while the full flat dispersion of the VHS1 can be observed at other photon energies (54 eV, 108 eV, corresponding to $k_z = 0$ as same as 78 eV), indicating that the diminished intensity of the flat-band top at 78 eV (Fig.3e) is attributed to the matrix elements effect (see Supplementary Fig. S7 for details).*".

REVIEWERS' COMMENTS

Reviewer #2 (Remarks to the Author):

With the new data on the k_z dispersion of VHSs at M in the latest revision, as well as the photon energy dependence data near the zone center added in previous resubmission, the current manuscript provides a detailed measurements on the electronic structure in this novel Kagome superconductor. While the significance of the higher-order VHS1 would be debatable due to moderate k_z dispersion, the experimental data focusing on VHSs are quite comprehensive and of decent quality, hence deserve the publication in Nature Communications.

Point-to-Point Response to Reviewers' Reports

For clarity, the reviewers' original comments are shown by blue italic characters.

The authors' responses are shown by black normal characters.

Reviewer #2 (Remarks to the Author):

*With the new data on the k_z dispersion of VHSs at M in the latest revision, as well as the photon energy dependence data near the zone center added in previous resubmission, the current manuscript provides a detailed measurements on the electronic structure in this novel Kagome superconductor. While the significance of the higher-order VHS1 would be debatable due to moderate k_z dispersion, the experimental data focusing on VHSs are quite comprehensive and of decent quality, hence deserve the publication in *Nature Communications*.*

We thank Reviewer #2 for reviewing our paper again and recommending its publication in *Nature Communications*.